# Exploration of Gender Differences in COVID-19 Discourse on Reddit

**Jai Aggarwal**        **Ella Rabinovich**        **Suzanne Stevenson**

Department of Computer Science, University of Toronto

{jai,ella,suzanne}@cs.toronto.edu

## Abstract

Decades of research on differences in the language of men and women have established postulates about the nature of lexical, topical, and emotional preferences between the two genders, along with their sociological underpinnings. Using a novel dataset of male and female linguistic productions collected from a social media platform, we further confirm existing assumptions about gender-linked affective distinctions, and demonstrate that these distinctions are amplified in the emotionally-charged discourse related to COVID-19. Our analysis also reveals considerable differences in topical preferences between male and female authors in pandemic-related discussions.

## 1   Introduction

Research on gender differences in language has a long history spanning psychology, gender studies, sociolinguistics, and, more recently, computational linguistics. A considerable body of linguistic studies highlights the differences between the language of men and women in topical, lexical, and syntactic perspectives (Lakoff, 1973; Labov, 1990); these differences have proven to be accurately detectable by automatic classification tools (Koppel et al., 2002; Schler et al., 2006; Schwartz et al., 2013). Here, we study the differences in male (M) and female (F) language in discussions of COVID-19[1] on the Reddit[2] discussion platform. Responses to the virus on social media have been heavily emotionally-charged, accompanied by feelings of anxiety, grief, and concern regarding long-lasting effects, such as economic ones. We explore how established emotional and topical cross-gender distinctions are carried over into pandemic-related discourse.

Multiple studies (e.g., Mulac et al. (2001); Mulac (2006); Newman et al. (2008)) have found distinctions in *topical preferences* in spontaneous productions of the two genders, showing that men were more likely to discuss money- and occupation-related topics, while women preferred discussion on family and social life. The authors attributed the differences to the assumption that male authors are more likely to discuss objects and impersonal topics, while female authors are more interested in psychological and social processes.

Gender-linked linguistic distinctions across *emotional dimensions* have been a subject of prolific research, both from the perspective of comprehension and production (Burriss et al., 2007; Hoffman, 2008; Thelwall et al., 2010), with findings suggesting that women are more likely than men to employ positive emotions, while men exhibit higher tendency to dominance, engagement, and control (although see Park et al. (2016) for an alternative finding). A common way to study emotions in psycholinguistics uses an approach that groups affective states into a few major dimensions. The Valence-Arousal-Dominance (VAD) affect representation has been widely used to conceptualize an individual's emotional spectrum, where *valence* refers to the degree of positiveness of the affect, *arousal* to the degree of its intensity, and *dominance* represents the level of control (Bradley and Lang, 1994). Computational studies applying this approach to emotion analysis have been relatively scarce due to the limited availability of a comprehensive resource of VAD rankings, with (to the best of our knowledge) no large-scale study on cross-gender language. The NRC-VAD Lexicon, a large dataset of VAD human rankings, recently released by Mohammad (2018), facilitates computational analysis of gender-linked differences across the three emotional dimensions at scale.

We use the VAD dataset of Mohammad (2018) to perform a comprehensive analysis of the similarities and differences between M and F language

---

[1]We refer to COVID-19 by 'COVID' hereafter.
[2]https://www.reddit.com/

collected from the Reddit discussion platform, contrasting two sub-corpora: a collection of spontaneous utterances on a wide variety of topics (the 'baseline' dataset), and a collection of COVID-related productions by the same set of authors. We first corroborate existing assumptions on differences in emotional aspects of linguistic productions of men and women, and further show that these distinctions are amplified in the emotionally-intensive setting of COVID discussions. We next take a topic modeling approach to show detectable distinctions in the range of topics discussed by the two genders in COVID-related discourse, reinforcing (to some extent) assumptions on gender-related topical preferences, in emotionally-charged discourse.[3]

## 2 Datasets

Our main dataset comprises a large collection of spontaneous, COVID-related English utterances by male and female authors from the Reddit discussion platforms. As of May 2020, Reddit was ranked as the 19th most visited website in the world, with over 430M active users, 1.2M topical threads (subreddits), and over 70% of its user base coming from English-speaking countries. Subreddits often encourage their subscribers to specify a meta-property (called a 'flair', a textual tag), projecting a small glimpse about themselves (e.g., political association, country of origin, age), thereby customizing their presence within a subreddit.

We identified a set of subreddits, such as 'r/askmen', 'r/askwomen', where authors commonly self-report their gender[4], and extracted a set of unique user-ids of authors who specified their gender as a flair. Using the extracted set of ids along with their associated gender, we collected COVID-related submissions and comments[5] by $10,421$ male and $5,630$ female users from the Reddit discussion platform, starting February 1st through June 1st, resulting in over 70K male and 35K female posts spanning $7,583$ topical threads. COVID-related posts were identified by matching a set of predefined keywords with a post's content: 'covid', 'covid-19', 'covid19', 'corona', 'coronavirus', 'the virus', 'pandemic'. The ample size

of the corpus facilitates analysis of distinctions—along emotional and topical dimensions—between the two genders in their discourse on the pandemic. Figure 1 presents the weekly amount of COVID-related posts in our main corpus. As can be seen, the discourse was increased in early-mid March (weeks 5–6), followed by a gradual decrease in intensity until nearly flattening out during the last four weeks of our analysis.

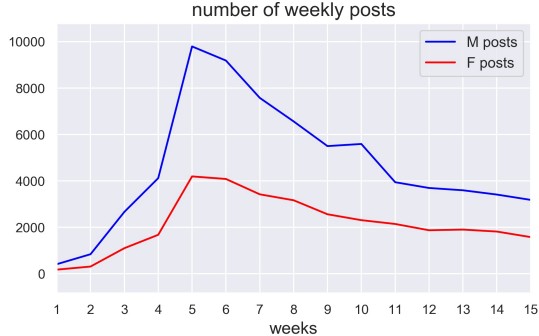

Figure 1: Weekly amount of posts by gender.

Aiming at a comparative analysis between virus-related and 'neutral' (baseline) linguistic productions by men and women, we collected an additional dataset comprising randomly sampled 10K posts per week by the same set of authors, totalling in 150K posts for each gender. We use the collected data for analysis of emotional differences as well as topical preferences in spontaneous productions by male and female authors on Reddit.

## 3 Analysis of Emotional Dimensions

### 3.1 Methods

A large dataset of VAD human rankings for $20,000$ English words has been recently released by Mohammad (2018), where each word is assigned V, A, and D values, each in the range [0–1]. For example, the word 'fabulous' is ranked high on the valence dimension, while 'deceptive' is rated with a low score. In this study we aim at estimating the affective variables of posts (typically comprising multiple sentences), rather than individual words; we do so by inferring the affective rankings of sentences using those of individual words.

Word embedding spaces have been shown to capture variability in emotional dimensions closely corresponding to valence, arousal, and dominance (Hollis and Westbury, 2016), implying that such semantic representations carry over information useful for the task of emotional affect assessment. Therefore, we exploit affective dimension ratings

---

[3]All data and code will be available at https://github.com/ellarabi/covid19-demography.

[4]Although gender can be viewed as a continuum rather than binary, we limit this study to the two most prominent gender markers in our corpus: male and female.

[5]For convenience, we refer to both initial submissions and comments to submissions as 'posts' hereafter.

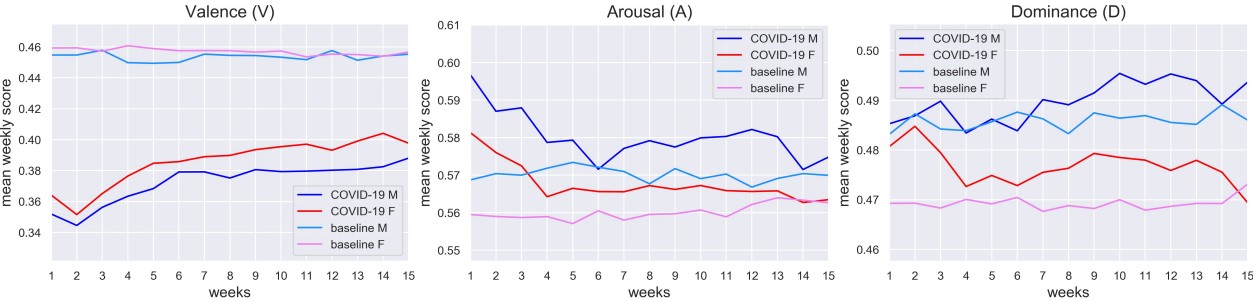

Figure 2: Diachronic analysis of valence (left), arousal (middle), and dominance (right) scores for Reddit data.

| | COVID-related posts | | | | | baseline posts | | | | |
|---|---|---|---|---|---|---|---|---|---|---|
| | mean(M) | std(M) | mean(F) | std(F) | eff. size | mean(M) | std(M) | mean(F) | std(F) | eff. size |
| V | 0.375 | 0.12 | **0.388** | 0.11 | -0.120 | 0.453 | 0.14 | **0.459** | 0.14 | -0.043 |
| A | **0.579** | 0.09 | 0.567 | 0.08 | 0.144 | **0.570** | 0.10 | 0.559 | 0.09 | 0.109 |
| D | **0.490** | 0.08 | 0.476 | 0.07 | 0.183 | **0.486** | 0.09 | 0.469 | 0.09 | 0.185 |

Table 1: Comparison of M and F means for each affective dimension. All differences are significant at p<0.001. The highest mean score in a row (for COVID and baseline data, separately) is boldfaced.

assigned to individual words for supervision in extracting ratings of sentences. We use the model introduced by Reimers and Gurevych (2019) for producing word- and sentence-embeddings using Siamese BERT-Networks,[6] thereby obtaining semantic representations for the 20, 000 words in Mohammad (2018) as well as for sentences posted by Reddit authors. This model performs significantly better than alternatives (such as averaging over a sentence's individual word embeddings and using BERT encoding (Reimers and Gurevych, 2019)) on the SentEval toolkit, a popular evaluation toolkit for sentence embeddings (Conneau and Kiela, 2018).

Next, we trained beta regression models[7] (Zeileis et al., 2010) to predict VAD scores (dependent variables) of words from their embeddings (independent predictors), yielding Pearson's correlations of 0.85, 0.78, and 0.81 on a 1000-word held-out set for V, A, and D, respectively. The trained models were then used to infer VAD values for each sentence within a post using the sentence embeddings.[8] A post's final score was computed as the average of the predicted scores for each of its constituent sentences. As an example, the post *'most countries handled the covid-19 situation appropriately'* was assigned a low arousal score of 0.274, whereas a high arousal score of 0.882 was assigned to *'gonna shoot the virus to death!'*.

---

[6]We used the `bert-large-nli-mean-tokens` model, obtaining highest scores on a the STS benchmark.

[7]An alternative to linear regression in cases where the dependent variable is a proportion (in 0–1 range).

[8]We excluded sentences shorter than 5 tokens.

### 3.2 Results and Discussion

We compared V, A, and D scores of M posts to those of F posts, in each of the COVID and baseline datasets, using Wilcoxon rank-sum tests. All differences were significant, and Cohen's d (Cohen, 2013) was used to find the effect size of these differences; see Table 1. We also compared the scores for each gender in the COVID dataset to their respective scores in the baseline dataset (discussed below). We further show, in Figure 2, the diachronic trends in VAD for M and F authors in the two sub-corpora: COVID and baseline.

First, Table 1 shows considerable differences between M and F authors in the baseline dataset for all three emotional dimensions (albeit a tiny effect size in valence), in line with established assumptions in this field (Burriss et al., 2007; Hoffman, 2008; Thelwall et al., 2010): women tend to use more positive language, while men score higher on arousal and dominance. Interestingly, the cross-gender differences in V and A are amplified between baseline and COVID data, with an increase in effect size from 0.043 to 0.120 for V and 0.109 to 0.144 for A. Men seem to use more negative language when discussing COVID than women do, presumably indicating a grimmer outlook towards the pandemic outbreak. Virtually no difference was detected in D between M and F authors in baseline vs. virus-related discussions.

COVID-related data trends (Figure 2) show comparatively low scores for valence and high scores for arousal in the early weeks our analysis (Febru-

| topics with highest coherence scores in M posts | | | | topics with highest coherence scores in F posts | | | |
|---|---|---|---|---|---|---|---|
| M-1 | M-2 | M-3 | M-4 | F-1 | F-2 | F-3 | F-4 |
| money | week | case | fuck | virus | feel | mask | week |
| economy | health | rate | mask | make | thing | hand | test |
| business | close | spread | claim | good | good | wear | hospital |
| market | food | hospital | news | thing | friend | woman | sick |
| crisis | open | week | post | vaccine | talk | food | patient |
| make | travel | month | comment | point | make | face | symptom |
| economic | supply | testing | call | happen | love | call | doctor |
| pandemic | store | social | article | human | parent | store | positive |
| lose | stay | lockdown | chinese | body | anxiety | close | start |
| vote | plan | measure | medium | study | read | stay | care |

Table 2: Most coherent topics identified in M and F COVID-related posts.

ary to mid-March). We attribute these findings to an increased level of alarm and uncertainty about the pandemic in its early stages, which gradually attenuated as the population learned more about the virus. Intuitively, both genders exhibit lower V scores in COVID discussions compared to baseline: Cohen's $d$ effect size resulted in $-0.617$ for M and $-0.554$ for F authors. Smaller, yet considerable, differences between the two sub-corpora exist also for A and D (0.095 and 0.047 for M, as well as 0.083 and 0.085, for F authors). Collectively, these affective divergences from baseline typify emotionally-intensive COVID-related discourse.

## 4   Analysis of Topical Distinctions

We next explored detailed topical similarities and differences in the productions by the two genders. Specifically, we compared two topic models: one created using M posts, and another using F posts, in the COVID dataset.[9] We identified the prevalent discussion topics in these two sub-corpora by using a publicly-available topic modeling tool (MALLET, McCallum, 2002). Each topic is represented by a probability distribution over the entire vocabulary, where terms more characteristic of a topic are assigned a higher probability. A common way to evaluate a topic learned from a set of documents is by computing its *coherence score* – a measure reflecting mutual semantic similarity of the topic's terms, and, therefore, its overall quality (Newman et al., 2010). The quality of a learned model is then estimated by averaging the scores of its individual topics – the *model* coherence score. We selected the optimal number of topics for each set of posts

by maximizing its model coherence score, resulting in 8 topics for male and 7 topics for female posts (coherence scores of 0.48 and 0.46).

We examined the similarities and the differences across the two topical distributions by extracting the top-4 topics – those with the highest individual coherence scores – in each of the M and F models. Table 2 presents the 10 words with highest likelihood for these topics in each model (on the left and right sides, respectively); topics within each are ordered by decreasing coherence score (left to right). We can see that both genders are occupied with health-related issues (topics M-3, F-1, F-4), and the implications on consumption habits (topics M-2, F-3). However, clear distinctions in topical preferences are also revealed by our analysis: men discuss economy/market and media-related topics (M-1, M-4), while women focus more on family and social aspects (F-2). Collectively these results show that the established postulates regarding gender-linked topical preferences are evident in COVID-related discourse on Reddit.

## 5   Conclusions

A large body of studies spanning a range of disciplines has suggested (and corroborated) assumptions regarding the differences in linguistic productions of male and female speakers. Using a large dataset of COVID-related utterances by men and women on the Reddit discussion platforms, we show clear distinctions along emotional dimensions between the two genders, and demonstrate that these differences are amplified in emotionally-intensive discourse on the pandemic. Our analysis of topic modeling further highlights distinctions in topical preferences between men and women.

---

[9]Prior to topic modeling we applied a preprocessing step including lemmatization of a post's text and filtering out stop-words (the 300 most frequent words in the corpus).

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
