# OpenReview forum: "Exploration of Gender Differences in COVID-19 Discourse on Reddit"
_aclweb.org/ACL/2020/Workshop/NLP-COVID — NLP-COVID-2020_

### Official Review · AnonReviewer2 · 2020-06-26
**Overall the paper is well written, contains re-usable data, and describes clear results.**

**Rating:** 7
**Confidence:** 2

**Review:**

Quality:
Overall the paper is well written, contains re-usable data, and describes clear results.

Clarity:
Authors aims of analysis were clearly stated, as were the methods employed. Results are elucidated clearly. The paper is well written, concise, and easy to follow logically.

Originality:
Given the findings corroborate already established patterns of F / M speech, the exact findings that those patterns persist in covid-related speech is not particularly original. However, within the context of studying phenomena amidst a completely novel world event, covid, the findings regarding how people talk about said event are original. Combination of methodologies to perform analysis are somewhat original.

Significance:
Mohammed's VAD showed low inter-annotator agreement for A & D types. This may reduce the impact of any findings, distinctions, or variances (even if statistically significant) between the genders in these categories. Even if statistically significant, the cohen-d effect sizes between F & M are still very small (< .2 in all categories). What is the _human significance_ (not mathematical significance) of the analyzed differences?

Editing suggestions:
  Clarify Fig1 caption by including "re covid" or something to that effect
  Typo in sentence, missing "of" :  "COVID-related data trends (Figure 2) show comparatively low scores for valence and high scores
for arousal in the early weeks [OF] our analysis (February to mid-March)".
  This sentence comes off as sexist: "women tend to use more positive language, while men score higher on arousal and dominance." Use similar terms to describe characteristics for both genders instead of saying what women do and what men score, e.g, "Women score higher in use of positive language, while men score ..."

pros
  Straightforward, solid results that established F / M speech patterns persist in novel corpus.
  Probably a decent baseline paper to use in further research on gender differences re covid speech or other domains.

cons
  Statistical significance does not explain human importance of findings.

---

> ### Author Response · Authors · 2020-06-26
> **Response to Reviewer**
>
> Thank you for taking the time to read and review our paper, we appreciate the feedback. To address your points:
>
> 1.   We appreciate your three editing suggestions, and will implement
> them for the camera-ready version.
>
> 2.   Regarding human importance: We agree that further work could
> improve our understanding of the differences in the arousal and dominance
> dimensions. In particular, a potential route might be to collect annotations on
> a sample of the posts to see whether the differences we detect with our
> approach, while small, are also perceptible by human raters. A more elaborate method
> might be to see whether comments to posts with automatically-detectable differences
> in emotional intensity are influenced by these differences; such an approach
> would require careful establishment of controls. In any case, given the short
> timelines, these considerations were not addressable in the current paper.
>
> Thank you again for your
> feedback.

---

### Official Review · AnonReviewer3 · 2020-07-03
**Overall the paper is okay but fails to provide the significance of the work.**

**Rating:** 4
**Confidence:** 4

**Review:**

This paper aims to understand the difference between male and female discourse in social media looking at a manually annotated set of Reddit threads related to Covid-19 compared to a baseline set. They confirm existing results about male and female discourse on the VAD scale.

The paper is clear and well-written and seems to be an interesting analysis, but fails to provide the significance of the work. Further the only novelty of the work is the application to Covid-19, otherwise all methods are utilizing previous work. This is not to say the authors should re-invent the wheel.


Pros:
- An interesting exploration of gender differences that confirms previous results.
- A good use of previous work on a new corpus.

Cons:
- Missing the overall significance for researchers, clinicians, epidemiologists, etc.
- It is unclear why Reddit specifically is used. They mention it is the 19th most visited site world. What about the other ones that are more visited? Is Reddit truly representative of the population at large? A description of the basic characteristics of Reddit users and posts would be helpful.
- There is also a large imbalance between male and female posts (2:1 ratio).
- This is very heteronormative.
- The dataset is pulled from 15 weeks starting from Feb 1 to June 1, which was a rapidly changing time. The paper would benefit from a discussion of the different topics discussed over that time in comparison to the topics pulled out by the models. Currently we are in a new "normal" and I think that would be reflected during the different weeks.
- The baseline is pulled from the same time period of Covid-19. An explanation of why the baseline should be the same time frame would be helpful to understand why the baseline is not from before Covid-19 when males and females were posting "normal" stuff.
- The overall results in table 1 are confusing in what is being compared and what is statistically significant. The difference between males and females for the VAD criteria may be statistically significant but it is a minor increase (< 0.2). It is unclear how important this is and what implications it has.
- A more in depth discussion on the relevance of the most coherent topics for males and females would be helpful.

---

> ### Author Response · Authors · 2020-07-04
> **Response to Reviewer**
>
> Dear reviewer,
>
> Thank you for your useful feedback! Please find below our response addressing your points.
>
> - We see the significance of our findings in demonstrating that emotional and topical differences in gender-linked language (shown thus far on small amounts of data and/or number of authors) are borne out and amplified in an emotionally-intensive setting. The novelty of this work lies in applying a novel combination of existing techniques (VAD ratings of words + regression models applied on word and sentence embeddings) for inferring emotional ratings of multi-sentence submissions. We will emphasize these contributions in the camera-ready version, if the paper is accepted.
>
> - The Reddit discussion platform is one of the most popular social media outlets for NLP research. Its monthly traffic closely follows that of twitter and facebook, with the number of active users comparable to that of twitter (330M as of 2020): it represents at large the population active on social media. Offering unlimited length of posts, and a convenient way for providing meta-data about users' traits (such as demographics), this platform provides an improved ability to detect emotion and topics in natural text (compared to character-limited twitter messages). Additionally, our work relies on accurate, self-reported gender of tens of thousands of authors -- information not easily obtainable from other social media outlets.
>
> - Indeed, male authors are more active on reddit (or, at least, they more commonly self-report their gender). As we collected utterances of over 16,000 authors comprising over 100,000 posts, we believe that this uneven distribution does not introduce any statistical bias into our analysis.
>
> - We agree that work on gender divided only into male and female language varieties is not ideal, but unfortunately we are limited by two factors: (1) most work in linguistics and psycholinguistics has focused on these dominant genders, and (2) the vast majority of users on reddit who self-report their gender do so choosing either male or female. An advantage of working on reddit (as opposed to pursuing traditional experimental or case-study analyses) is that, as the amount of text grows and the number of users grows who self-report genders other than male and female, eventually there should be sufficient samples of language that would enable going beyond the traditional male/female categories.
>
> - We conducted topic modeling analysis on the set of posts discussing the pandemic. A diachronic analysis in the dynamics of topics (unrelated to the virus) being discussed on social media and reflecting the new "normal" would definitely be an interesting direction for future work.
>
> - We selected our baseline data from the same time period in order to control for other time-dependent factors that could influence the language samples, so that the only difference was whether the posts mentioned the COVID-related keywords or not. We thought this was the most conservative approach (ie, findings could not be linked to other changes between, say, a year ago and the past few months). With over 50,000 daily submissions across a wide range of topical threads (subreddits), we believe that our baseline data (collected from a parallel timeline) reliably reflect general, pandemic-unrelated ("normal") linguistic utterances across the two genders. To dispel any doubt, we can perform a similar comparison to baseline data collected from a period preceding the virus outbreak for the camera-ready version.
>
> - All differences in Table 1 are statistically significant (as mentioned in the caption). The low effect sizes reflect the (expectedly) small, yet non-trivial and reliably detected, differences between male and female speakers. We are unaware of prior work reporting cross-gender differences in VAD dimensions (for meaningful comparison), but low effect sizes are not an uncommon finding in the psycholinguistic literature. More importantly, we show that these divergences are amplified in an emotionally-charged setup, which is one of the main contributions of this study.
>
> - We will extend the discussion on topic modeling results for the camera-ready version.

---

### Official Review · AnonReviewer1 · 2020-07-05
**nice application to new data set to be made available**

**Rating:** 7
**Confidence:** 3

**Review:**

This paper explores gender differences in linguistic productions between two groups of Redditors who self-identify as either "male" or "female". It examines a corpus of covid-19 pandemic threads in relation to two areas: emotion analysis (employing a VAD lexicon and word embedding representations); and topic analysis (employing the tool MALLET).

The paper's novelty is in the application of an established method to a new corpus that the authors have developed pertaining to covid-19 threads. As expected, the language usage for covid-19 posts had a lower Valence when compared to language used in a baseline corpus. There is also a general trend for the language used in the female sub-corpus scoring slightly higher in the Valence scale than male sub-corpus. The trends are reversed when Arousal and Dominance were examined: overall higher for men, and when comparing baseline to the covid-19 posts, the baselines score slightly lower for both male and female data.

To compare and contrast the different topics covered between the male and female authored posts, topic modelling was applied to each sub-corpus and the topics with the highest coherence scores were presented. However, applying topic modelling to the corpus as a whole and analysing the topic allocation of the male and female posts would give a better indication of similarities and differences of the topics covered in the sub-corpora. However, two different topic models for each sub-corpus were developed and the most cohesive topics were presented.

In general the VAD study is interesting, although unsurprsing. The goal of discovering if different or similar topics were covered in the two sub-corpora may be best approached by discovering the topics covered by the corpus as a whole and analysing the topic allocation of the sub-corpora.

---

> ### Author Response · Authors · 2020-07-06
> **Reply to Reviewer**
>
> Thank you for taking the time to read and review our paper, we appreciate the feedback!
>
> Performing topic modelling on the corpus as a whole indeed seems like a strong alternative to our methodology. If time permits, we can conduct this analysis and include it in the camera-ready version of the paper.

---

### Decision · Program_Chairs · 2020-07-06

**Decision:**

Accept

**Comment:**

Thank you for your submission.

The reviewers have identified that this is a well-written paper presenting interesting results over a new data set. While there are a number of issues raised, we are satisfied by the response from the authors and believe that the concerns can be addressed in a revision for the final proceedings (which we will provide instructions about after the workshop).

We therefore welcome your presentation at the workshop on Thursday (5:30-9:30pm PDT). Please plan on a 10-minute presentation, pre-recorded or live.

Sincere apologies for the lateness of the decision!

---

> ### Author Response · Authors · 2020-07-06
> **Reply to Decision**
>
> Thank you! Looking forward to presenting this Thursday.